# Diffusion denoising as a certified defense against clean-label poisoning attacks

## Abstract

We present a certified defense to clean-label poisoning attacks. These attacks work by injecting poisoning samples that contain $p$-norm bounded adversarial perturbations into the training data to induce a targeted misclassification of a test-time input. Inspired by the adversarial robustness achieved by *denoised smoothing*, we show how a pre-trained diffusion model can sanitize the training data before a model training. We extensively test our defense against seven clean-label poisoning attacks and reduce their attack success to 0–16% with only a small drop in the test time accuracy. We compare our defense with existing countermeasures against clean-label poisoning, showing that the defense reduces the attack success the most and offers the best model utility. Our results highlight the need for future work on developing stronger clean-label attacks and using our certified yet practical defense as a strong baseline to evaluate these attacks.

## 1 Introduction

A common practice in machine learning is to train models on a large corpus of data. This paradigm empowers many over-parameterized deep-learning models but also makes it challenging to retain the quality of data samples collected from various sources. This makes deep-learning-enabled systems vulnerable to *data poisoning* (Rubinstein et al., 2009; Nelson et al., 2008b; Biggio et al., 2012; Jagielski et al., 2018; Shafahi et al., 2018; Carlini, 2021)—where an adversary can modify a victim model's behaviors by injecting malicious samples (i.e., poisons) into the training data.

But in practice, it may be challenging to make modifications to a training datasets used for fully-supervised classification, because they are often relatively small. As a result, recent work has developed *clean-label poisoning attacks* (Shafahi et al., 2018; Zhu et al., 2019; Aghakhani et al., 2021; Turner et al., 2019; Saha et al., 2020; Huang et al., 2020; Geiping et al., 2021b), where the attacker aims to control the victim model's behavior on a few specific test inputs by injecting poisons that visually appear to be correctly-labeled, but in fact include human-invisible adversarial perturbations.

We introduce a defense against targeted clean-label poisoning attacks inspired by defenses to test-time adversarial examples. Existing poisoning defenses fall into two categories *certified* and *heuristic*. Certified defenses offer provable guarantees, but often significantly decrease the utility of defended models at test-time, making them impractical (Ma et al., 2019; Levine & Feizi, 2020; Wang et al., 2022; Zhang et al., 2022). Heuristic approaches (Suciu et al., 2018; Peri et al., 2020; Hong et al., 2020; Geiping et al., 2021a; Liu et al., 2022) demonstrate their effectiveness against existing attacks in realistic scenarios. However, these defenses rely on unrealistic assumptions, such as the defender knowing the target test-time input (Suciu et al., 2018), or are evaluated against specific poisoning adversaries, leaving them improvable against adaptive attacks or future adversaries.

**Contributions.** We *first* make two seemingly distant goals closer (i.e., providing a provable defense that *also* works in realistic settings) and present a certified defense against clean-label poisoning. Our defense offers a provable guarantee while minimizing the decrease in clean accuracy. For any $p$-norm bounded adversarial perturbations to the training data, we ensure a certified accuracy higher than the prior certified defenses. The model trained on the tampered data classifies a subset of test-time input $x$ (or $x + \delta$, where $||\delta||_{\ell_p}$ in clean-label backdoor poisoning) correctly. To achieve this goal, we leverage the recent diffusion probabilistic diffusion models (Sohl-Dickstein et al., 2015; Ho et al., 2020; Nichol & Dhariwal, 2021). We use an off-the-shelf diffusion model to *denoise*

the entire training data before training a model. In §3, we theoretically show how one achieves a certified accuracy under 2-norm bounded adversarial perturbation to the training data.

Removing adversarial perturbations in the training data before model training, we can decouple the certification process from the training algorithm. *Second*, we leverage this computational property of our defense and present a series of training techniques to alleviate the side-effects of employing certified defenses, i.e., the utility loss of a defended model. To our knowledge, we are the first work that decouples these two processes. We propose the warm-starting (Ash & Adams, 2020)–training a model a few epochs on the tampered data–and to initialize the model parameters using a pre-trained model. In §4, we show they attribute to a better performance compared to existing certified defenses.

*Third*, we extensively evaluate our defense against seven clean-label poisoning attacks in two different scenarios: training a model from-scratch and transfer-learning. In transfer-learning, our defense reduces the attack success to 0% with a negligible accuracy drop of 0.5% in the certified radius of 0.1 in 2-norm perturbations. We also reduce the attack success to 2–16% when a defender trains a model from scratch. We further compare our defense with six poisoning defenses in the prior work. We show more reduction in the attack success and less accuracy drop than those existing defenses.

*Fourth*, in §5, we discuss research questions that are important in studying poisoning attacks and defenses but are far-neglected in the prior work. We suggest future work directions to answer them.

## 2 PRELIMINARIES ON CLEAN-LABEL POISONING

**Threat model.** A clean-label poisoning attacker causes the targeted misclassification of a specific *target* test-time sample $(\mathbf{x}_t, y_t)$ by compromising the training data $D_{tr}$ with clean-label poisoning samples $D_p$. If a victim trains a model $f$ on the poisoned training data $(D_{tr} \cup D_p)$, the resulting model $f_\theta^*$ is likely to misclassify the target instance to the adversarial class $y_{adv}$ while preserving the classification behavior of $f_\theta^*$ the same on the clean test-set $S$. The attacker crafts those *poisons* $(\mathbf{x}_p, y_p) \in D_p$ by first taking a few *base* samples in the same domain $(\mathbf{x}_b, y_b)$ and then adding human-imperceptible perturbations $\delta$, carefully crafted by the attacker and also bound to $||\delta||_{\ell_p} \leq \epsilon$, to them while keeping their labels intact $(y_b = y_p)$. A typical choice of the bound is $\ell_2$ or $\ell_\infty$.

**Poisoning as a constrained bi-level optimization.** The process of crafting optimal poisoning samples $(D_p^*)$ can be formulated as the constrained bi-level optimization problem:

$$D_p^* = \underset{D_p}{\arg\min}\, \mathcal{L}_{adv}(x_t, y_t; f_\theta^*),$$

where $\mathcal{L}_{adv}(x_t, y_{adv}; f_\theta^*)$ is the adversarial loss function quantifying how accurately a model $f_\theta^*$, trained on the compromised training data, misclassifies a target sample $x_t$ into the class an adversary wants $y_{adv}$. $D_p$ is the set of poisoning samples we craft, and $D_p^*$ is the resulting optimal poisons.

While minimizing the crafting objective $\mathcal{L}_{adv}(x_t, y_t; f_\theta^*)$, the attacker also trains a model $f_\theta^*$ on the compromised training data, which is itself another optimization problem, formulated as follows:

$$f_\theta^* = \underset{\theta}{\arg\min}\, \mathcal{L}_{tr}(D_{tr} \cup D_p, S; \theta),$$

where the typical choice of $\mathcal{L}_{tr}$ is the cross-entropy loss, and $S$ is the clean test-set. Combining both the equations becomes a bi-level optimization: find $D_p$ such that $\mathcal{L}_{adv}$ is minimized after training.

To make the attack inconspicuous, the attacker *constraints* this bi-level optimization by limiting the amount of perturbation $\delta = ||x_p - x_b||$ each poisoning sample can contain to $||\delta|| < \epsilon$.

**Existing poisoning attacks.** Initial work (Shafahi et al., 2018; Zhu et al., 2019; Aghakhani et al., 2021) minimizes $\mathcal{L}_{adv}$ by crafting poisoning samples that are close to the target in the latent representation space $g(\cdot)$. A typical choice of $g(\cdot)$ is the activation outputs from the penultimate layer of a pre-trained model $f(\cdot)$, publicly available to the attacker from the Internet. The attacks have shown effectiveness in *transfer-learning* scenarios, where $f(\cdot)$ will be fine-tuned on the poisoned training data. The attacker chooses base samples from the target class $(x_b, y_{adv})$ and craft poisons $(x_p, y_{adv})$. During fine-tuning, $f(\cdot)$ learns to correctly classify poisons in the target's proximity in the latent representation space, which will classify the target into the class $(y_{adv})$ the attacker wants.

Recent work focuses on making those poisoning attacks effective when $f$ is trained *from scratch* on the compromised training set. To craft poisoning samples effective in "from scratch" scenarios,

the attacker requires to approximate the gradients computed on $\mathcal{L}_{adv}$ that are likely to appear in any models. Huang et al. (2020) addresses this challenge by meta-learning; the poison-crafting process simulates all the possible initialization, intermediate models, and adversarial losses computed on those intermediate models. A follow-up work by Geiping et al. (2021b) alleviates the computational overhead of the prior work by proposing an approximation technique, gradient matching, that aligns the gradients from poisoning samples with those computed on a target.

**Defenses against clean-label poisoning.** Early work on poisoning defenses focuses on filtering out poisons from the training data. Nelson et al. (2008a) and Suciu et al. (2018) compute the training samples negatively impacting the classification results on targets. But in practice, the defender does not know which test-time samples are the attack targets. Follow-up work (Rubinstein et al., 2009; Peri et al., 2020; Tran et al., 2018) leverages unique statistical properties (*e.g.*, spectral signatures) that can distinguish poisons from the clean data. All these approaches depend on the dataset they use to compute the properties. Recent work (Geiping et al., 2021a; Borgnia et al., 2021; Hong et al., 2020; Liu et al., 2022) thus reduces the dependency by proposing data- or model-agnostic defenses, adapting robust training or differentially-private training to poisoning defenses. While shown effective against existing attacks, those defenses are not *provable*.

A separate line of work addresses this issue by proposing *certifiable* defenses (Levine & Feizi, 2020; Wang et al., 2022; Weber et al., 2023), which guarantee the correct classification of test-time samples when an adversary compromises the training data with the number of poisons $k$ and their perturbation radius of $r$. (Levine & Feizi, 2020; Wang et al., 2022)'s approach is majority voting, where the defender splits the training data into multiple disjoint subsets, train models on them, and run majority voting of test inputs over those models. (Zhang et al., 2022) proposes BagFlip, a model-agnostic defense that leverages the idea of randomized smoothing studied in adversarial robustness. However, the certification offered by these works has only shown effectiveness against toy datasets like MNIST, while the effectiveness is unknown against practical poisoning attacks, such as clean-label poisoning in CIFAR10. It further hinders the deployment of such defenses as they are computationally expensive to derive such certifications.

Our work proposes a provable yet computationally efficient defense against clean-label poisoning that leverages diffusion-denoising probabilistic models. We will show the certification our defense can offer (§3) and the defense outperforms empirical defenses against existing attacks (§4).

# 3 A PRACTICAL, CERTIFIED DEFENSE USING DIFFUSION DENOISING

**Problem definition.** We are interested in certifying the prediction of a model $f_\theta^*$ trained on the tampered training data $(D_{tr} \cup D_p)$. To achieve this goal, we first define the perturbation radius $r$ as the $\ell_2$ distance between two datasets, computed by taking the sum across $\ell_2$ perturbations of all images. The perturbation space $P_r^\pi$ is the set of all the poisoning datasets $D_p$ obtained by tampering samples with the crafting algorithm $\pi$ with the perturbation radius of $r$. Operating within this perturbation space $P$, for any given test-time sample $(\mathbf{x}_t, y_t)$, can produce a certificate that

$$Pr_{D_p \in P_r^\pi(D_{tr})} \left[ f_{\theta \leftarrow D_p}^*(\mathbf{x}_t) = f_{\theta \leftarrow D_{tr}}^*(\mathbf{x}_t) \right] > 1 - \alpha$$

with arbitrarily high probability $1 - \alpha$, we guarantee that training on any poisoned dataset $D_p \in P_r^\pi(D_{tr})$ in the perturbation radius $r$ will classify the example $\mathbf{x}_t$ the same way as it was classified on the non-poisoned dataset.

## 3.1 DENOISING DIFFUSION PROBABILISTIC MODEL

In our work, we use off-the-shelf denoising diffusion probabilistic model (DDPMs) (Sohl-Dickstein et al., 2015; Ho et al., 2020; Nichol & Dhariwal, 2021) to remove adversarial perturbations added to the training data by the clean-label poisoning attacker.

DDPMs are a recent generative model that works by learning the diffusion process of the form $x_t \sim \sqrt{1 - \beta_t} \cdot x_{t-1} + \beta_t \cdot \omega_t$, where $\omega_t$ is drawn from a standard normal distribution $\mathcal{N}(0, \mathbf{I})$ with $x_0$ sourced from the actual data distribution, and $\beta_t$ being fixed (or learned) variance parameters. The above process transforms images from the target data distribution to complete random noise over time, and the reverse process creates images from the data distribution starting with random

Gaussian noise. In image classification, a diffusion model with a fixed time-step $t \in \mathbb{N}^+$ and a fixed schedule samples a noisy version of a training image $x_t \in [-1, 1]^{w \cdot h \cdot c}$ of the form:

$$x_t := \sqrt{\alpha_t} \cdot x + \sqrt{1 - \alpha_t} \cdot \mathcal{N}(0, \mathbf{I}),$$

where $\alpha_t$ is a constant derived from $t$, which decides the level of noise to be added to the image (the noise increases consistently as $t$ grows). During training, the model minimizes the difference between $x$ and the denoised version of $x_t$, where $x_t$ is obtained by applying the noise at time-step $t$.

## 3.2 PREDICTION AND CERTIFICATION

Now we present practical algorithms for running a prediction with the classifiers trained on a poisoned training set $D_{tr} \cup D_p$, and certifying the robustness of the prediction of a test-time sample **x**. Our work extends the algorithms in prior work on *randomized smoothing* (Cohen et al., 2019) to clean-label poisoning settings. Our algorithms are shown in Algorithm 1.

We first train many classifiers on noised-then-denoised training datasets, and then return the predictions of these classifiers on the target example $x_t$, with the algorithm `NoiseTrainClassify`. Then, we follow exactly the standard Randomized Smoothing procedure and produce a certificate with `Certify`, which yields a radius on which this example $x_t$ is guaranteed to be robust.

---

**Pseudocode** Noise, Denoise, Train, Classify

```
1: NoiseTrainClassify(D, f, x, σ)
2:     t*, α_{t*} ← GetTStep(σ)
3:     # run denoised smoothing on x and D
4:     D_{t*}, x_{t*} ← Noise(D, x, σ)
5:     D̂_{t*}, x̂ ← Denoise(D_{t*}, x_{t*}; t*)
6:     f_{θ_{t*}} ← Train(D̂_{t*}, f)
7:     return f_{θ_{t*}}(x̂)
8:
9: GetTStep(σ)
10:    t* ← find t s.t. (1−α_t)/α_t = σ²
11:    return t*, α_{t*}
```

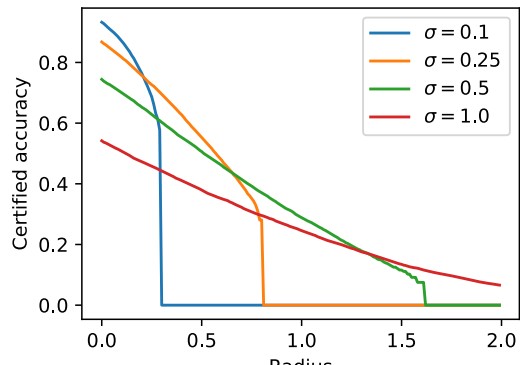

Figure 1: **Prediction and certification** can be implemented with a few (∼10) lines of code.

Figure 2: **Certified radius and accuracy** for differing levels of Gaussian noise $\sigma$ added during training.

---

**Prediction.** To compute the prediction for a test-time example, we exactly follow standard randomized smoothing and run `NoiseTrainClassify` algorithm a sufficient number of times (e.g., 1000) and to compute independent predictions for different noised training datasets. The output is then the majority vote label from these classifiers we train.

**Certification.** Producing a certificate also exactly follows standard randomized smoothing: we first count the number of occurrences of the most likely label compared to any other runner-up label, and from this can derive a radius on which this point is guaranteed to be robust.

**Results.** We train $10,000$ CIFAR10 models using an efficient training pipeline that trains a single model in 15 seconds on an A100 GPU (for a total of 1.7 GPU days of compute). These models have a clean accuracy of between $92\%$ and $58\%$ depending on the level of noise introduced, and for the first time can certify nontrivial robustness to clean label poisoning attacks on CIFAR-10. A full curve comparing certified accuracy for all given perturbation radii is shown in Figure 2.

## 3.3 COMPUTATIONAL EFFICIENCY FOR CERTIFICATION

In order to obtain a robustness certificate, we must train many CIFAR10 models on different denoisings of the training dataset. This step of the process is computationally expensive, but is in line with the computational complexity of prior randomized smoothing based approaches that require similar work. However, as we will show later, in practice we can empirically obtain robustness with only a very small number of training runs (even as low as **one**!), confirming the related observation from (Lucas et al., 2023) who find a similar effect holds for randomized smoothing of classifiers.

## 4 EMPIRICAL EVALUATION

Now we empirically evaluate the effectiveness of our certified defense against clean-label poisoning attacks. We utilize the poisoning benchmark developed by Schwarzschild et al. (2021). This benchmarks run clean-label poisoning attacks with the same attack and training configurations, which allows us to compare our defense's effectiveness across multiple attackers.

**Poisoning attacks.** We evaluate our defense against seven clean-label poisoning attacks: four targeted poisoning (Poison Frogs! (Shafahi et al., 2018), Convex Polytope (Zhu et al., 2019), Bullseye Polytope (Aghakhani et al., 2021), Witches' Brew (Geiping et al., 2021b)) and three backdoor attacks (Label-consistent Backdoor (Turner et al., 2019), Hidden-Trigger Backdoor (Saha et al., 2020), and Sleeper Agent (Souri et al., 2022)). The first three targeted poisoning attacks operate in the transfer-learning scenarios, whereas the rest assumes a victim's model being trained from-scratch. We note that while our defense can offer provable guarantees only to targeted poisoning attacks, we want to test whether the defense can mitigate clean-label backdooring like the prior work did.

**Metrics.** We consider two metrics: clean accuracy and attack success rate. We compute the accuracy on the entire test set. We consider an attack successful when a poisoned model classifies a target sample to the intended class. When evaluating backdoor attacks, we exclude the cases from the attack success when a tampered model misclassifies the target sample without any trigger pattern.

**Methodology.** We run our experiments in CIFAR10 (Krizhevsky et al., 2009). Following the standard practices in the prior work (Schwarzschild et al., 2021), we run each attack 100 times on different targets and report the averaged metrics over the 100 runs. We randomly choose 10 targets from each of the 10 CIFAR10 classes. We set the poisoning budget to 1% for all our attacks, e.g., the attacker can tamper with only 500 samples of the entire 50k training instances in CIFAR10.

We evaluate against both the transfer-learning and training from-scratch attacks. In transfer-learning, we fine-tune a pre-trained model on the tampered training set while we train a randomly initialized model in the training from-scratch scenario. We also evaluate our defense against the attackers with two types of knowledge on a victim: white-box and black-box. The white-box attackers know the victim model's architecture (and its parameters), while the black-box attackers do not.

We describe the attack configurations and training hyper-parameters in detail in Appendix A.

### 4.1 DIFFUSION DENOISING RENDERS CLEAN-LABEL POISONING INEFFECTIVE

We first show to what extent our defense makes the seven poisoning attacks ineffective. Our defense is attack-agnostic. A defender does not know whether the training data contains poisoning samples or not; instead, the defender always trains a model on the denoised training set. We measure the trained model's accuracy and attack success and compare them with the baseline, where a model is trained directly without denoising. We evaluate with different defense strengths: $\sigma \in \{0.1, 0.25, 0.5, 1.0\}$. The results are from attacking with the perturbation bound of 16-pixels ($||\delta||_\infty = 16$). Our results against a weaker poisoning attack ($||\delta||_\infty = 8$) are shown in Appendix B.

| Poisoning attacks | $K$ | Scenario | Our denoising defense at $\sigma$ (%) | | | | |
|---|---|---|---|---|---|---|---|
| | | | $^\dagger$**0.0** | **0.1** | **0.25** | **0.5** | **1.0** |
| Witches' Brew (Geiping et al., 2021b) | 1 | WB | $^{(92.2)}$44.0 | $^{(86.5)}$6.0 | $^{(72.4)}$4.0 | $^{(46.5)}$10.9 | $^{(42.1)}$12.1 |
| | 4 | | $^{(92.3)}$65.0 | $^{(86.5)}$9.0 | $^{(71.9)}$3.0 | $^{(46.0)}$9.0 | $^{(41.3)}$7.0 |
| Sleeper Agent (Souri et al., 2022) | 4 | | $^{(92.2)}$31.5 | $^{(86.0)}$15.7 | $^{(72.8)}$10.0 | $^{(46.5)}$30.5 | $^{(42.9)}$14.1 |
| Witches' Brew (Geiping et al., 2021b) | 1 | BB | $^{(90.1)}$22.0 | $^{(84.6)}$8.1 | $^{(70.4)}$4.0 | $^{(49.2)}$9.9 | $^{(42.6)}$10.3 |
| | 4 | | $^{(90.0)}$33.5 | $^{(84.6)}$2.0 | $^{(70.1)}$2.0 | $^{(49.2)}$8.0 | $^{(42.7)}$9.0 |
| Sleeper Agent (Souri et al., 2022) | 4 | | $^{(91.0)}$16.9 | $^{(84.5)}$13.1 | $^{(69.6)}$10.6 | $^{(48.4)}$28.2 | $^{(42.7)}$11.1 |

Table 1: **Defense effectiveness in training from-scratch scenarios.** We measure the accuracy and attack success of the models trained on the denoised training set. Each cell shows the accuracy in the parentheses and the attack success outside. $^\dagger$ indicates the baseline (no-defense), $K$ is the number of ensembles, and WB and BB stand for the white-box and the black-box attacks, respectively.

**Training from-scratch scenarios.** Table 1 summarizes our results. Each cell contains the accuracy and the attack success rate of a model trained on the denoised training set. All the numbers are the averaged values over 100 attacks. We demonstrate that our defense mitigates both attacks. In Witches' Brew, the attack success decreases from 44–65% to 6–9%; against Sleeper Agent, the attack success decreases from 32% to 13% at most[1]. We also show that to reduce their attack success to ∼10%, our defense needs at a minimum $\sigma$ of 0.1–0.25. Increasing $\sigma$ may defeat a stronger attack but significantly decrease a model's accuracy. This result is interesting, as in our theoretical analysis shown in Sec 3, our defense requires $\sigma$ of 0.25, while 0.1 is sufficient to defeat both the attacks.

| Poisoning attacks | Scenario | Our denoising defense at $\sigma$ (%) | | | | |
|---|---|---|---|---|---|---|
| | | [†]0.0 | 0.1 | 0.25 | 0.5 | 1.0 |
| Poison Frog! (Shafahi et al., 2018) | White-box | (93.6)69.5 | (93.3)0.0 | (92.6)0.0 | (90.8)0.0 | (87.5)0.0 |
| Convex Polytope (Zhu et al., 2019) | | (93.7)12.6 | (93.3)0.0 | (92.6)0.0 | (90.8)1.0 | (87.6)0.0 |
| Bullseye Polytope (Aghakhani et al., 2021) | | (93.5)100 | (93.3)0.0 | (92.6)0.0 | (90.8)0.0 | (87.5)0.0 |
| Label-consistent Backdoor (Turner et al., 2019) | | (93.4)6.3 | (93.3)0.0 | (92.6)0.0 | (90.7)0.0 | (87.4)0.0 |
| Hidden Trigger Backdoor (Saha et al., 2020) | | (93.2)1.0 | (93.3)0.0 | (92.6)0.0 | (90.8)0.0 | (87.4)0.0 |
| Poison Frog! (Shafahi et al., 2018) | Black-box | (92.2)4.2 | (91.8)0.0 | (90.8)0.0 | (89.3)0.0 | (86.7)0.0 |
| Convex Polytope (Zhu et al., 2019) | | (92.3)3.1 | (91.8)0.0 | (90.8)0.0 | (89.3)0.0 | (86.7)0.0 |
| Bullseye Polytope (Aghakhani et al., 2021) | | (92.0)11.7 | (91.8)0.0 | (90.8)0.0 | (89.3)0.0 | (86.7)0.0 |
| Label-consistent Backdoor (Turner et al., 2019) | | (92.1)6.3 | (91.8)0.0 | (90.8)0.0 | (89.3)0.0 | (86.7)1.9 |
| Hidden Trigger Backdoor (Saha et al., 2020) | | (91.5)1.0 | (91.8)0.0 | (90.8)0.0 | (89.3)0.0 | (86.7)1.4 |

Table 2: **Defense effectiveness in transfer-learning scenarios.** We measure the clean accuracy and attack success of the models trained on the denoised training set. Each cell shows the accuracy in the parentheses and the attack success outside. Note that [†] indicates the baseline without any denoising.

**Transfer-learning scenarios.** Table 2 shows our results with the same format as in Table 1. Our defense is also effective in mitigating the attacks in transfer-learning. Against the white-box attacks, we reduce the success rate of the targeted poisoning attacks by 28–90% at $\sigma$ of 0.1. In the black-box attacks, the targeted poisoning is less effective (with 3–12% attack success), but still, the defense reduces the success to 0%. We observe that the two clean-label backdoor attacks show 1–6% success regardless of the attacker's knowledge. We therefore exclude these two attacks from the subsequent evaluation. Using $\sigma$ greater than 0.5 increases the attack success slightly. However, this is not because of the successful poisoning attacks but due to the significant decrease in a model's accuracy (5-7%). Here we had the same results: defeating these poisoning attacks requires much smaller $\sigma$ values (i.e., $\sigma = 0.1$) than what we've seen in our theoretical analysis ($\sigma = 0.25$).

## 4.2 IMPROVING MODEL UTILITY UNDER OUR DENOISING DEFENSE

A shortcoming of existing certified defenses (Levine & Feizi, 2020; Wang et al., 2022; Zhang et al., 2022) is that the certified accuracy is substantially lower than the undefended model's accuracy. We also observe in the previous subsection that our defense, when we train a model from scratch on the denoised data, is not free from the same issue, which hinders their deployment in practice.

The theoretical analysis in §3 shows that our defense is agnostic to how we *initialize* a model. We leverage this property and minimize the utility loss by initializing model parameters in specific ways before we train the model on the denoised data. We test two scenarios: a defender can use in-domain data or out-of-domain data to pre-train a model and use its parameters to initialize.

**Initializing model parameters using in-domain data.** Our strategy here is an adaptation of warm-starting (Ash & Adams, 2020). We first train a model from scratch on the tampered training data for a few epochs to achieve high accuracy. It only requires 5–10 epochs in CIFAR10. We then apply our defense and continue training the model on the denoised training set. We evaluate this strategy against the most effective attacks of each type: Bullseye Polytope (targeted attacks in transfer-learning), Witches' Brew (targeted attacks in training from-scratch), and Sleeper Agent (backdoors).

Table 3 summarizes our results. The middle three rows are the results of leveraging the in-domain data. We train the ResNet18 models on the tampered training set for 10 epochs and continue training

---

[1]Note that the typical attack success rate of backdoor attacks are ∼90%; Sleeper Agent is a very *weak* attack.

| Poisoning attacks | Pre-training data | Model | Warm-start | Our denoising defense at $\epsilon$ (%) | | | | |
|---|---|---|---|---|---|---|---|---|
| | | | | [†]0.0 | 0.1 | 0.25 | 0.5 | 1.0 |
| Bullseye Polytope
Witches' Brew
Sleeper Agent | N/A
(Baseline) | ResNet18 | ✗ | (93.5)100.0
(92.3)65.0
(92.2)31.5 | (93.3)0.0
(86.5)9.0
(86.0)15.7 | (92.6)0.0
(71.9)3.0
(72.8)10.0 | (90.8)0.0
(46.0)9.0
(46.5)30.5 | (87.5)0.0
(41.3)7.0
(42.9)14.1 |
| Bullseye Polytope
Witches' Brew
Sleeper Agent | In-domain Data
(CIFAR10, Tampered) | ResNet18 | ✓ | (93.5)100.0
(92.2)65.0
(92.2)31.5 | (93.3)0.0
(86.5)9.0
(86.0)18.6 | (92.6)1.0
(**72.7**)5.0
(**73.3**)8.8 | (**90.9**)1.0
(**47.2**)7.0
(**48.1**)9.6 | (**87.7**)2.0
(38.0)10.0
(36.0)11.5 |
| Bullseye Polytope
Witches' Brew
Sleeper Agent | Out-of-domain Data
(ImageNet-21k, Clean) | [†]ResNet18 | ✓ | (84.4)8.0
(92.3)65.0
(92.4)33.2 | (83.4)2.0
(83.4)2.0
(**86.4**)14.0 | (78.4)0.0
(**78.4**)0.0
(72.7)5.4 | (67.5)0.0
(67.5)6.0
(46.8)10.1 | (54.5)6.0
(**54.5**)6.0
(38.6)11.2 |

[†]ResNet18 in Torchvision library; only the latent space dimension differs from the ResNet18 in prior work.

Table 3: **Improving the performance of defended models.** We show the accuracy of defended models and the attack success rate when we employ two techniques for initializing their parameters. The top three rows are the baseline results from Table 1 and 2, and the next two sets of three rows show our results. We highlight the cells showing the accuracy improvements in **bold**.

on the denoised version of the training data for the rest 30 epochs. We show that our strategy (warm-starting) can increase the accuracy of the defended models while defeating clean-label poisoning attacks. Under strong defense guarantees $\sigma > 0.1$, the models have 0.5–2.2% increased accuracy, compared to the baseline, while keeping the attack success ~10%. In $\sigma = 0.1$, we achieve the same accuracy and defense successes. Our strategy can be potentially useful when a defender needs a stronger guarantee, such as against stronger clean-label poisoning future work will develop.

**Using models pre-trained on out-of-domain data.** Now, instead of running warm-starting on our end, we can also leverage "warm" models available from the legitimate repositories. To evaluate, we take the ResNet18 model pre-trained on ImageNet-21k[2]. We use this pre-trained model in two practical ways. We take the model as-is and train it on the denoised training data. In the second approach, we combine the first approach with the previous idea. We train the model on the tampered training data for a few epochs and then train this fine-tuned model on the denoised data.

The bottom three rows of Table 3 are the results of using the warm model. We fine-tune the ResNet18 for 40 epochs on the denoised training data. Surprisingly, our second strategy can further improve the accuracy of the final models trained with strong defense guarantees (e.g., $\sigma > 0.1$). The final models achieve 6.5–13.2% greater accuracy than the results shown in Table 1 and 2. In $\sigma = 1.0$, the final accuracy of defended models has a negligible difference from the baseline. We are the first work to offer practical strategies to manage the utility-security trade-off in certified defense.

### 4.3 COMPARISON TO EXISTING POISONING DEFENSES

We finally examine how our defense works better/worse than existing poisoning defenses. We compare ours with six state-of-the-art defenses: DP (Ma et al., 2019; Hong et al., 2020), DPA (Levine & Feizi, 2020; Wang et al., 2022), BagFlip (Zhang et al., 2022), ROE (Rezaei et al., 2023), k-NN (Peri et al., 2020), AP (Geiping et al., 2021a), and FrieNDs (Liu et al., 2022). The first four are certified defenses like ours, and the last three are non-certified ones. Here we test them in the same attack settings we use in the previous sections (e.g., $||\delta||_\infty = 16$ and in CIFAR10).

We first start with comparing our defense with approaches to filtering out poisoning samples.

| Poisoning attacks | TP | FP | k-NN | Ours |
|---|---|---|---|---|
| Witches' Brew | 31 | 464 | (86.3)3.0 | (86.5)9.0 |
| Bullseye Polytope | 431 | 63 | (93.7)24.2 | (93.3)0.0 |

Table 4: **Comparison to Deep k-NN.** Our defense is better than Deep k-NN (Peri et al., 2020) at mitigating clean-label poisoning attacks.

**Comparison to Deep k-NN.** Peri et al. (2020) remove poisons from the training data by leveraging the k-nearest-neighbor (k-NN) algorithm. They run k-NN on the latent representations of the training data to identify potentially malicious samples (i.e., samples with labels different from the nearest ones) and remove them. We compare Deep k-NN's effectiveness to our defense. We use the defense configurations that bring the best results in the original study, i.e.,

[2]https://pytorch.org/vision/stable/models/generated/torchvision.models.resnet18.html

setting $k$ to 500. Table 5 summarizes our results. Deep k-NN fails to remove most poisoning samples (but it removes many benign samples!). Our defense achieves a higher success in defeating clean-label poisoning attacks.

Now, we turn our attention to the training-time defenses and compare ours with them.

| | Heuristic defense | | | Certified defense | |
|---|---|---|---|---|---|
| **Poisoning attacks** | **DP-SGD** | **AT** | **FrieNDs** | **ROE** | **Ours** |
| Bullseye Polytope (Aghakhani et al., 2021) | (93.9)7.0 | (90.3)96.4 | (90.2)10.0 | *N/A | (93.5)0.0 |
| Witches' Brew (Geiping et al., 2021b) | (76.4)4.0 | (66.3)2.4 | (87.6)8.0 | (70.2)10.0 | (86.5)9.0 |
| Sleeper Agent (Souri et al., 2022) | (74.7)2.3 | (66.3)7.2 | (87.4)13.6 | (68.6)12.0 | (86.0)15.7 |

*ROE trains 250 models from scratch on each partition of the training data; incompatible with BP.

Table 5: **Comparison to training-time defenses.** We compare with three heuristic defenses and one recent certified defense against clean-label poisoning attacks most effective in our experiments.

**vs. DP-SGD.** Ma et al. (2019) proposed a certified defense against poisoning attacks that leverage differential privacy (DP). They make a model less sensitive to a single-sample modification to the training data. But the defense offers a weak guarantee in practice; for example, the certificate is only valid in CIFAR10, when an adversary tampers one training sample. Hong et al. (2020) later empirically shows that DP is still somewhat effective against poisoning. We compare our defense to the training with $(1.0, 0.05)$-DP[3] on the same tampered training data. We use Opacus[4] to train models with DP and keep all the other configurations the same. In Table 5, both defenses significantly reduce the attack success to 0–16%. Our defense achieves at most 10% higher accuracy than DP. DP reduces the attack success slightly more against clean-label backdooring.

**vs. AT.** Geiping et al. (2021a) adapts the adversarial training (AT) (Madry et al., 2017) for defeating clean-label poisoning: in each mini-batch, instead of crafting adversarial examples, they synthesize poisoning samples and have a model to make correct classifications on them. While shown some effectiveness, this approach could overfit a specific poisoning attack considered in the adapted AT, leaving the model vulnerable to unseen poisoning attacks. We therefore compare our defense with the original AT with the PGD-7 bounded to $\epsilon$ of 4. Table 5 shows that our defense achieves higher accuracy than the robust models while making the attacks ineffective. Interestingly, AT cannot defeat the BP attacks. We hypothesize that clean-label poisons in the training data are already adversarial examples; thus, PGD-7 may not add more perturbations to these poisons during training.

**vs. FrieNDs.** Liu et al. (2022) advances the idea of training robust models. Instead of adding random Gaussian noise to the data during training, they use "friendly" noise, pre-computed by a defender, that minimizes the accuracy loss to a model they will train. When their defense is in action, they train a model on the tampered training data with two types of noise: the friendly noise computed before and a weak random noise sampled uniformly within a bound. We compare our defense with FrieNDs, and both greatly reduce the attack's success while preserving the model's utility. The critical difference is that ours is a certified defense, while FrieNDs is not. A promising future work we leave here is developing adaptive clean-label poisoning attacks against FrieNDs.

**vs. ROE.** Certified defenses against data poisoning attacks (Steinhardt et al., 2017; Ma et al., 2019; Diakonikolas et al., 2019; Levine & Feizi, 2020; Gao et al., 2021; Zhang et al., 2022; Wang et al., 2022; Rezaei et al., 2023) are; however, we found that most of these defenses showed their effectiveness in limited scenarios, such as in MINST-17 (Zhang et al., 2022), a subset of MNIST containing only the samples of the digits 1 and 7. In consequence, they aren't compatible with our clean-label poisoning settings. A recent defense by Rezaei et al. (2023) demonstrates the certified robustness in practical settings; we thus compare our defense to their run-off-election (ROE). In Table 5, we empirically show that our defense is more effective than ROE while minimizing the accuracy drop.

**Comparison of computational overhead.** Most training-time defenses require additional mechanisms we need to "add" to the training algorithm. AT needs to craft adversarial examples (or clean-label poisons (Geiping et al., 2021a)). DP-SGD introduces noise into the gradients during training. FrieNDs pre-computes friendly noise and then optimally applies the noise during the training, and

---

[3]We set the clipping norm and noise multiplier to 1.0 and 0.05, following the prior work (Lecuyer et al., 2019).
[4]https://opacus.ai

ROE requires training of 250 base classifiers to achieve the improved certificates. In contrast, our defense only requires to run forward *once* with an off-the-shelf diffusion model.

## 5 DISCUSSION AND FUTURE WORK

Our work offers a new perspective on the cat-and-mouse game played in clean-label poisoning.

**Initial claims from the cat:** Clean label poisoning attacks, with human-imperceptible perturbations that keeping the original labels, are an attractive option to inject malicious behaviors into models. Even if a victim carefully curates the training data, these attacks remain effective by evading *filtering* defenses that rely on statistical signatures distinguishing clean samples from poisons (Nelson et al., 2008a; Suciu et al., 2018; Peri et al., 2020).

**The mouse's counterarguments:** Yet, as seen in research on the adversarial robustness, such small perturbations can be *brittle* (Carlini et al., 2022): by adding a large quantity of noise and then denoising the images, the adversarial perturbations become nearly removed—and so clean-label perturbations are also removed. In prior work (and in this work as well), we leverage this observation and propose both certified and non-certified heuristic defenses.

**Additional arguments from the cat:** A counterargument from the clean-label attack's side was that those defenses inevitably compromise a model's performance—a fact corroborated by the prior work on the adversarial robustness (Tsipras et al., 2018; Zhang et al., 2019). Similarly, defenses could greatly reduce the poisoning attack success, but at the same time, they decrease the classification accuracy of defended models, often tipping the scales in favor of the adversary. If a defense yields a CIFAR-10 model with 60–70% utility, what is the point of having such models in practice? Indeed, our own certified models require degraded utility to achieve certified accuracy. And other defenses, e.g., differentially-private training (Ma et al., 2019; Hong et al., 2020), are computationally demanding, increasing the training time of a model by an order of magnitude.

Our work shows that this cat-and-mouse game need not be as pessimistic as the cat portrays.

By leveraging an off-the-shelf diffusion denoising model (Ho et al., 2020; Nichol & Dhariwal, 2021), we can purify the training data, possibly containing malicious samples, *offline* and render six existing clean-label attacks ineffective. With a small provable guarantee against the training data contamination ($||\epsilon||_\infty = 2$), the majority of attacks reach to 0% success. A few recent attacks exhibit 0–10% success rate in CIFAR10—a rate comparable to random misclassification of test-time inputs.

In contrast to the cat's counterargument, we can also address the accuracy degradation problem—as long as we do not need certified robustness. Existing defenses that work to defend against poisoning require applying a particular training algorithm designed by the defender (Hong et al., 2020; Geiping et al., 2021a; Liu et al., 2022; Levine & Feizi, 2020; Wang et al., 2022; Zhang et al., 2022). By decoupling the training process and the defense, we open a new opportunity for a defender to develop training techniques that can enhance the accuracy of the defended models while maintaining robustness. Our experimental results propose two effective heuristic algorithms in improving defended models' utility, especially when the models are under strong defense guarantees (i.e., $||\epsilon||_\infty > 2$).

**Next steps?** One of our main conclusions—other than that our defense is effective—is that there is an extremely large gap between the certified accuracy and best practical attack accuracy. When using diffusion models to denoise adversarial examples, a practical adversary can achieve within a few percentage points of the worst-case certified result; for example, if we can certify a lower bound of 50% robust accuracy, a practical adversary could achieve 55% robust accuracy. But in this setting, even at perturbation levels where we can't certify *anything*, the attacks fail to decrease the model utility at all. This implies either that (1) current attacks are *not fully exploiting* the space of possible attacks, or (2) that the certificate provided is loose and a more careful analysis could tighten it.

In addition, when studying stronger clean-label poisoning attacks, we encourage future work to include our defense as a strong baseline. We show in §4 that the defense renders the state-of-the-art clean-label attacks ineffective and is, by far, the most effective defense among the existing ones. We believe that strong attacks future work will develop should at least survive against this defense. Our defense is also designed to run without any additional training costs—with an off-the-shelf diffusion model, one can easily denoise the tampered training data and run the training of a model on it.

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
