## A  EXPERIMENTAL SETUP IN DETAIL

Our framework is an adaption of the poisoning benchmark from the prior work (Schwarzschild et al., 2021). Most recent work on clean-label poisoning attacks uses this benchmark for showcasing their attack success. We implemented our framework in PyTorch v1.13[5] and Python v3.7[6]. We use the exact attack configurations and training hyper-parameters that the original study uses.

We only made two differences. We first increase the perturbations bounded to $||\epsilon||_\infty = 16$ as the 8-pixel bound attacks do not result in a high attack success rate. Defeating 8-pixel bounded attacks is trivial for any poisoning defenses. Second, we do not use their fine-tuning subset, which only contains 2500 training samples and 25–50 poisons. It (as shown in the next subsections) leads to 60–70% accuracy on the clean CIFAR10 test-set, significantly lower than 80–90%, which can be trivially obtained with any models and training configurations. If a model trained on the contaminated training data misclassifies a target, it could be a mistake caused by a poorly performing model.

Running each attack on 100 different poisoning samples is computationally demanding. We examine 7 attacks; for each, we run 100 times of crafting and training/fine-tuning a model. We also examine 5 different denoising factor $\sigma$. In total, we 3500 poisoning attacks. To accommodate this computational overhead, we use four machines, each is equipped with 8 Nvidia GPUs.

## B  ADDITIONAL EXPERIMENTAL RESULTS

Here, we include the results from our detailed experiments, e.g., the sensitivity of our defense against a weaker adversary whose perturbation is bounded to $||\delta||_\infty = 8$. We first craft 25 poisoning samples on a ResNet18 pre-trained on CIFAR-100. We then fine-tune the model on a subset of the CIFAR-10 training data. The subset contains the first 250 images per class (totaling 2500 training samples).

| Poisoning attacks | Scenario | Our denoising defense at $\sigma$ (%) | | | | |
|---|---|---|---|---|---|---|
| | | [†]0.0 | 0.5 | 1.0 | 1.5 | 2.0 |
| Poison Frog! (Shafahi et al., 2018) | White-box | (69.8)13.0 | (55.9)8.0 | (43.1)4.0 | (34.7)9.0 | (26.9)11.0 |
| Convex Polytope (Zhu et al., 2019) | | (69.8)24.0 | (53.8)5.0 | (42.6)9.0 | (35.0)8.0 | (27.7)6.0 |
| Bullseye Polytope (Aghakhani et al., 2021) | | (69.4)100.0 | (55.8)10.0 | (42.8)8.0 | (35.0)9.0 | (27.4)9.0 |
| Label-consistent Backdoor (Turner et al., 2019) | | (69.8)2.0 | (55.9)3.0 | (42.7)5.0 | (34.9)8.0 | (27.4)12.0 |
| Hidden Trigger Backdoor (Saha et al., 2020) | | (69.8)5.0 | (55.9)3.0 | (42.7)10.0 | (35.2)9.0 | (27.3)9.0 |
| Poison Frog! (Shafahi et al., 2018) | Black-box | (67.9)7.0 | (53.8)6.0 | (43.3)2.5 | (35.3)8.0 | (28.8)6.5 |
| Convex Polytope (Zhu et al., 2019) | | (67.9)4.0 | (53.7)3.0 | (43.3)3.5 | (35.2)3.0 | (28.8)8.0 |
| Bullseye Polytope (Aghakhani et al., 2021) | | (67.7)8.0 | (53.8)17.5 | (43.2)16.5 | (35.3)7.5 | (28.6)8.5 |
| Label-consistent Backdoor (Turner et al., 2019) | | (67.9)3.5 | (53.8)2.0 | (43.5)4.5 | (35.2)2.5 | (29.0)8.0 |
| Hidden Trigger Backdoor (Saha et al., 2020) | | (67.9)7.5 | (53.7)2.0 | (43.3)7.0 | (35.2)10.5 | (28.7)8.5 |

Table 6: **Diffusion denoising against clean-label poisoning.** We remove the $l_\infty$-norm of 8 perturbations added by five poisoning attacks by running a single-step stable diffusion on the entire training set. In each cell, we show the average attack success over 100 runs and the average accuracy of models trained on the denoised data in the parenthesis. ([†] indicates the no-defense scenario.)

Table 6 summarizes our results against clean-label poisoning attacks with $l_\infty$-norm of 8, respectively. We consider five poisoning attacks in the white-box and black-box scenarios. In the white-box setting, we fine-tune the ResNet18 model (that we use for crafting poisons) on the poisoned training set. In the black-box setting, we fine-tune different models (VGG16 and MobileNetV2 models pre-trained on CIFAR-100) on the same poisoned training set. We use $\sigma$ in $\{0.5, 1.0, 1.5, 2.0\}$ for our single-step diffusion denoiser. $\sigma$=0.0 is the baseline without the defense.

**Diffusion denoising significantly reduces poisoning attack success.** We first show that denoising the poisoned training set can significantly reduce the poisoning attack success. The most successful attack in the white-box setting, Bullseye Polytope, achieves the attack success of 100% and 95% in $l_\infty$-norm of 8 pixels, but denoising with $\sigma$ of 0.5 can reduce those to 10% and 5%. Denoising reduces the attack success of Poison Frog! and Convex Polytope from 13-34% to 2-8% at $\sigma = 0.5$.

---

[5]https://pytorch.org
[6]https://www.python.org/

The two backdoor attacks (Label-consistent and Hidden trigger) exploiting clean-label poisoning are not successful in the benchmark setup (their success rate ranges from 2-7.5%). We therefore could not quantify our defense's effectiveness against them. We re-configure the benchmark setups for them to increase their success rate and show the effectiveness of diffusion denoising against them.

**Strong denoising reduces a model's utility.** We also observe that the increased $\sigma$ (strong denoising) can significantly reduce the utility of a model trained on the denoised training data. As we increase the $\sigma$ from 0.5 to 2.0, the fine-tuned model's accuracy leads to 56% to 27%. However, we show that with the small $\sigma$, our diffusion denoising can reduce the attack success significantly. We also show in our evaluation section that we achieve a high model's utility while keeping the same $\sigma$. We attribute the increased utility to recent model architectures, such as VisionTransformers, or to pre-training a model on a larger data corpus. We leave further investigation for future work.

**Misclassification vs. poisoning attack success.** Moreover, in a few cases, the poisoning success increases from 2–7% to 10–13% as we increase $\sigma$. We attribute this increase not to the attack being successful with a high $\sigma$ but to the poor performance of a model. For example, the accuracy of a model with $\sigma = 2.0$ is $\sim$27%, meaning that four out of five targets in a class can be misclassified.

## C (DENOISED) POISONING SAMPLES

| FC | CP | BP | HTBD | CLBD | WB | $\sigma$ |
|---|---|---|---|---|---|---|

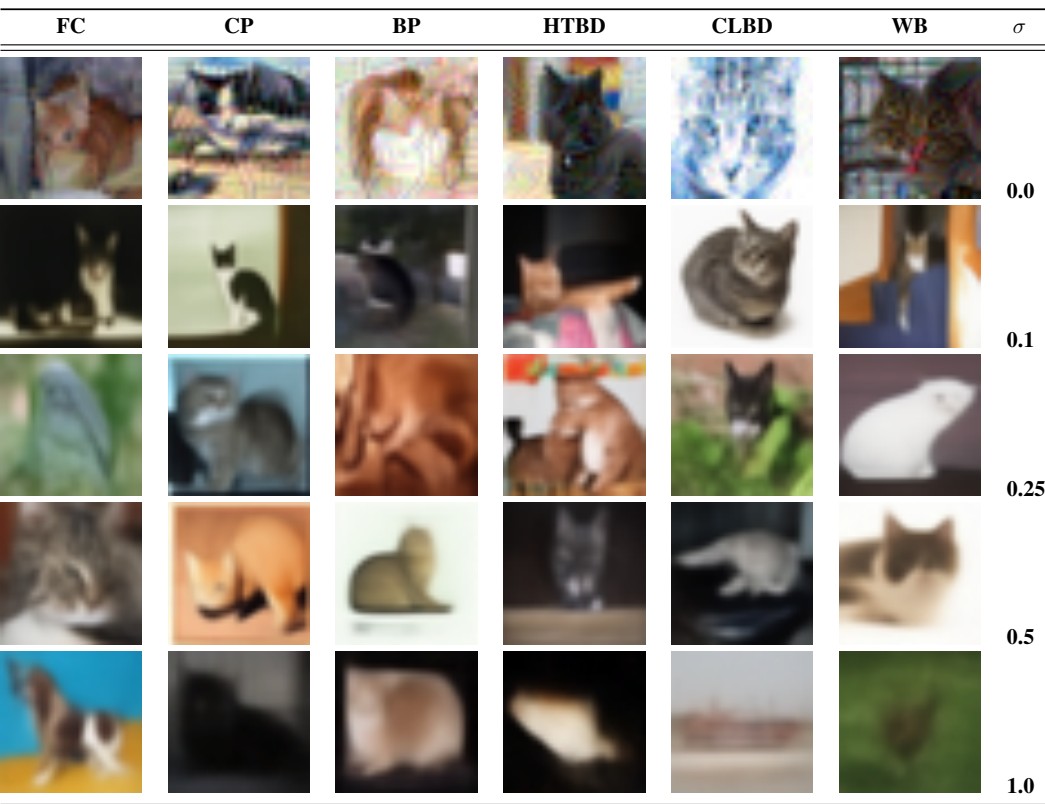

| | | | | | | 0.0 |
| | | | | | | 0.1 |
| | | | | | | 0.25 |
| | | | | | | 0.5 |
| | | | | | | 1.0 |

Table 7: **Visualize poisoning samples.** We, for the CIFAR10 training data, display the poisoning samples crafted by different clean-label poisoning attacks. We also show how the perturbations are *denoised* with difference $\sigma$ values in $\{0.1, 0.25, 0.5, 1.0\}$. $\sigma$=1.0 yields to ineffective poisons.