# OpenReview forum: "Diffusion Denoising as a Certified Defense Against Clean-Label Poisoning Attacks"
_ICLR.cc/2024/Conference — ICLR 2024 Conference Withdrawn Submission_

### Official Review · Reviewer_ehju · 2023-10-20

**Soundness:** 2 fair
**Presentation:** 1 poor
**Contribution:** 1 poor
**Rating:** 3
**Confidence:** 4

**Summary:**

In this paper, the authors propose a certified defense against clean-label data poisoning attacks using diffusion denoising. The authors sanitize the training dataset by using an off-the-shelf diffusion model to remove adversarial perturbations on poisoned data. They also propose a warm-starting initialization to improve the model utility under the defense. Experiments are carried out on CIFAR-10 to evaluate the defense against multiple attacks and comparisons are performed with existing heuristic and certified defenses.

**Strengths:**

1. The proposed approach is simple. The authors leverage the diffusion model to remove adversarial noise added on the poisoned data.

2. The authors proposed the strategy of warm-starting to improve the model utility.

3. The authors evaluate the proposed defense over difference settings such as training from scratch and transfer-learning.

**Weaknesses:**

1. One of the most important claims of this work is that the proposed approach is a certified defense and it offers a provable guarantee. However,  there is missing theoretical proof for the certified defense in the paper.

2. I favor the simplicity of the proposed approach (i.e., the authors use an off-the-shelf diffusion model to remove the noise imposed on the poisoned samples), but the motivation of this idea is not well explained. Why pick diffusion model as a denoising approach over other techniques? Why does diffusion model work well in removing the adversarial perturbations?

3. Experiments and evaluations are preformed only on CIFAR-10 dataset, which is insufficient to showcase the effectiveness and efficiency of the proposed attack. The authors mentioned that the proposed defense requires training of many models and is computationally expensive. Thus it is necessary to show the performance of the defense on more and large-scale datasets.

4. The comparison to existing defenses looks insufficient. Only three attacks and one certified defense are evaluated in Table 5. It would be desired to have a more comprehensive evaluation that includes more attacks and certified defenses.

5. The contribution of the proposed defense could be limited because it only considers the clean-label integrity attacks and backdoor attacks. Would it still be effective against clean-label availability attacks such as CLPA[1] or dirty-label backdoor attacks such as BadNets[2], LIRA[3], WaNet[4], etc? I suggest the authors better specify the scope of the work and clarify why other attacks are not considered.

6. This paper is not well polished. For example, the caption of Algorithm 1 is Figure 1 and there is no function called Certify in the algorithm. There are also typos and grammatical issues that makes the paper hard to follow.

[1]. CLPA: Clean-Label Poisoning Availability Attacks Using Generative Adversarial Nets, AAAI 2022

[2]. BadNets: Identifying Vulnerabilities in the Machine Learning Model Supply Chain

[3]. LIRA: Learnable, Imperceptible and Robust Backdoor Attacks, ICCV 2021

[4]. WaNet -- Imperceptible Warping-based Backdoor Attack, ICLR 2021

**Questions:**

Please see weaknesses for details.

---

### Official Review · Reviewer_iUfb · 2023-10-30

**Soundness:** 2 fair
**Presentation:** 1 poor
**Contribution:** 2 fair
**Rating:** 3
**Confidence:** 5

**Summary:**

This paper studies the adversarial robustness against the poisoning attack. The core idea of this paper is based on using diffusion denoising to improve the performance of randomized smoothing. In the randomized smoothing mechanism, inputs are injected by noise drawn from a noise distribution, both in the training and the inference. This training process can degrade the task performance due to the distortion of the injected noise. The diffusion model can be used as a denoiser to clean this noise and recover the semantic representation of the inputs. It starts with a noise variable and then denoises the variable continuously toward a sample from the normal data distribution. Therefore, this paper uses the diffusion model to denoise the noisy input in randomized smoothing, by considering the noisy input as the middle variable in the reverse diffusion process. Different from other works that improve the randomized smoothing using the diffusion model, this paper focuses on the certified robustness against poisoning attacks.

**Strengths:**

1. To the best of my knowledge, this is the first work that applies diffusion denoising to randomized smoothing against poisoning attack, which is a novel application.

2. The writing is easy to understand.

**Weaknesses:**

1. Some core algorithms/methodologies are missing. It is not clear how the randomized smoothing is applied to the poisoning attack. In Section 3.2, this paper states the certificate follows the standard randomized smoothing, which is confusing since the randomized smoothing is proposed to certify the robustness of evasion attacks. Since the poisoning attack shares a different setting with evasion attacks, the authors may want to clarify how the randomized smoothing guarantees the robustness against the poisoning attack. In addition, how the time step in the reverse diffusion process is computed according to $\sigma$ is not clear. The algorithm GetTStep is not well clarified in the main text. This is actually an important algorithm that is non-trivial.

2. No ablation study supports the main contribution. The main contribution is using the diffusion denoiser to improve the randomized smoothing, while this paper lacks the ablation study to compare the randomized smoothing performance w/ and w/o the diffusion denoising. It is not clear whether the diffusion denoising improves the randomized smoothing.

3. This paper claims that one of the contributions is to provide a provable defense that also works in realistic settings. This is actually controversial. When using the certified defense to defend against real attacks, the settings are very different and the results may be meaningless. For example, the randomized smoothing used in this paper provides the certified robustness in the criteria of $\ell_2$ norm radius, but the real attacks used in this paper adopt the $\ell_\infty$ constrain, so it is not fair to evaluate the $\ell_2$-norm defense against the $\ell_\infty$-norm attacks. In addition, the threat model could be very different in the empirical/certified defense. The model owner does not need to know the perturbation of the inputs in empirical defense, while the model owner needs to know the perturbation of the inputs when comparing the perturbation size to the radius, which is not realistic.

4. The application of randomized smoothing in poisoning attacks will raise a severe problem in running time. In this paper, to certify the perturbation to the training set on the prediction w.r.t. a specific sample, it requires training 10,000 models, plus the computation on the multi-step diffusion process, even for the fine-tuning setting, the running time overhead will be extremely large.

**Questions:**

1. Can you provide some quantitative results on the overhead of the robust training and the diffusion process?

2. How do you compute the step T in the diffusion process?

3. How does randomized smoothing guarantee robustness against poisoning attacks?

4. Can you provide the ablation study on randomized smoothing w/ and w/o denoising?

---

### Official Review · Reviewer_sFAv · 2023-10-31

**Soundness:** 2 fair
**Presentation:** 3 good
**Contribution:** 2 fair
**Rating:** 3
**Confidence:** 4

**Summary:**

This paper introduces a certified defense mechanism designed to counter clean-label poisoning attacks. These attacks involve injecting malicious samples into the training data, which carry adversarial perturbations bounded by a specified p-norm. These perturbations aim to manipulate the model's predictions for certain test inputs intentionally. Drawing inspiration from the concept of adversarial robustness achieved through denoised smoothing, the authors demonstrate how a pre-trained diffusion model can effectively cleanse the training data before model training.

The defense undergoes extensive testing against seven distinct clean-label poisoning attacks, significantly reducing their success rates to a range of 0–16%. This defense accomplishes this without a significant decrease in test-time accuracy. In comparative evaluations against existing countermeasures for clean-label poisoning, the method stands out by delivering the most substantial reduction in attack success while maintaining high model utility.

These results underscore the importance of future research efforts in developing more potent clean-label attacks. Moreover,the certified yet practical defense serves as a robust baseline for evaluating these attacks and enhancing the security of machine learning models.

**Strengths:**

1. The paper introduces a defense method based on DDPM to counter clean-label poisoning attacks, and it demonstrates strong performance against various attack methods.

**Weaknesses:**

1. The primary results presented in Table 1 are obtained through empirical experiments rather than relying on certified defense methods that guarantee a specific radius where the model is proven to provide the correct predictions. I would suggest a mathematical formulation to provide a certified bound.

2. Randomized smoothing has already been established as an effective technique for ensuring l2 or l_infinite robustness against various types of perturbations, including backdoor perturbations. Moreover, it offers a clear and established framework for calculating certified l2 or l_infinite robustness bounds. The novelty of this paper is somewhat limited, as its core idea revolves around utilizing randomized smoothing to provide certified defense specifically against backdoor perturbations.

3. Experiments on the indiscriminate data poisoning [1] should also be included.

[1] Yiwei Lu, Gautam Kamath, and Yaoliang Yu. 2023. Exploring the limits of model-targeted indiscriminate data poisoning attacks. In Proceedings of the 40th International Conference on Machine Learning (ICML'23), Vol. 202. JMLR.org, Article 949, 22856–22879.

**Questions:**

See weakness above.

---

### Official Review · Reviewer_mhLZ · 2023-11-02

**Soundness:** 2 fair
**Presentation:** 2 fair
**Contribution:** 2 fair
**Rating:** 3
**Confidence:** 3

**Summary:**

The paper proposes a certified defense against _clean label poisoning attacks_, where an attacker is allowed to modify each sample in the training dataset by $\epsilon$ in an $\ell_p$ norm in order to obtain a misclassification on a target test point $x$. The defense is a direct application of Randomized Smoothing to this problem, and proceeds by first adding gaussian noise to each image in the dataset, then denoising each image by a diffusion model, and finally training a classifier on this denoised dataset. The majority vote on $x$ over multiple classifiers obtained from such noise-denoise steps gives rise to a certifiably robust prediction.

**Strengths:**

1.	Denoising has been popular recently as a defense against adversarial examples, and this paper extends these works to the poisoning setting, which is interesting.

2.	The paper contains a nice overview of the field related to poisoning attacks, which is very helpful in contextualizing this work.

**Weaknesses:**

The paper consists of two parts: a robustness certificate (on the amount of poisoning adversarial corruption allowed per image in the dataset), and an empirical evaluation of the method with respect to known attacks in the literature.

1.	The presentation of the first part is unclear, missing key technical details

2.	The implementation of the second part raises several questions, pertinent to proper evaluation of defenses with random components in general, and defense using diffusion models in particular.

**Questions:**

1.	The authors should clarify the following technical details:
	1.	Is it right to understand that just as in traditional RS, we have $g(x) = {\rm argmax}_k Pr_v[f(x + v) = k]$, in this paper we have $g(x) = {\rm argmax}_k \Pr_v[{\rm Alg}_x(D + v) = k]$, where ${\rm Alg}_x$ is an algorithm where the test point $x$ is fixed, and it produces a classification for $x$ by examining the dataset $D$. In this paper, Alg corresponds to (Denoise D) -> Train classifier on D -> output prediction on $x$.   However, this procedure should produce an $\ell_2$ certificate on the entirety of $D$, which is unstructured, i.e., there is no restriction that the perturbation made to each image is $\ell_2$ bounded. This is different from an element-wise certificate on each image in $D$, which was the goal presumably.
	2.	Now, in the previous point, if the norm used is $\ell_\infty$, then it doesn’t matter because $\ell_\infty$ decomposes into individual pixels. However, now traditional RS certificates do not hold for $\ell_\infty$, and other noise distributions are more suitable [1].
	3.	In light of the above, it is extremely unclear what does “and from this we can derive a radius…” mean, and for that matter what is Figure 2 plotting. A derivation of the certificate (even if it is a reproduction of Cohen et. al.) obtained would make the statements clearer.

2.	Empirical Evaluation: One needs to be very careful while backpropagating gradients through diffusion models to attack them. Accordingly, the optimization steps in many of the tested poisoning attacks which explicitly use gradients of the networks are doomed to fail from the start. In particular:
	1.	See Section 3.2 in [1] for a detailed evaluation of how diffusion denoising can be broken as an adversarial defense, and what can be used to improve its real robust performance. Specifically, how does incorporating the techniques used in [2] (gradient estimation) to the optimization steps of all the tested attacks affect the evaluation?
	2.	More generally, adversarial evaluation of defenses with random components is very tricky [3], and typically the gradients need to be stabilized with EoT attacks as a first step towards proper evaluation. Accordingly, it is important to understand how do all the evaluated attacks perform when the gradients of the defense are stabilized.
	3.	The empirical evaluation seems to be on $\ell_\infty$ attacks, but the certificates were presumably on $\ell_2$ attacks. Gaussian smoothing is used to obtain certificates against $\ell_2$ attacks. For $\ell_\infty$ attacks, gaussian smoothing is not the best noise model, perhaps the results can be improved further by using the correct noise model.
	4.	The defense performance seems to shoot up even at very small magnitude of the injected noise $\sigma$. A reasonable conjecture would be that $\sigma > 0$ is not even needed for the empirical performances observed currently. In particular, the $\sigma = 0$ column should have the title “no defense”, and a separate $\sigma = 0$ column should be added which measures performance where no noise is added to the image, and it is simply passed through the forward and the backward process of a diffusion model. If the performance is indeed bad, then there needs to be an experiment finding exactly at what $\sigma$ does the defense start performing well, in other words, where is the jump from bad defense performance to good defense performance?
	5.	Related to the above, this shooting up of empirical defense performance is a tell-tale sign of gradient masking [3], and the empirical evaluation should include replacement of the optimization steps in all the poisoning defenses studied with a black-box gradient estimation oracle.

[1]: Randomized Smoothing of All Shapes and Sizes, Greg Yang, Tony Duan, J. Edward Hu, Hadi Salman, Ilya Razenshteyn, Jerry Li

[2]: Robust Evaluation of Diffusion-Based Adversarial Purification. Minjong Lee, Dongwoo Kim

[3]: On Evaluating Adversarial Robustness. Nicholas Carlini, Anish Athalye, Nicolas Papernot, Wieland Brendel, Jonas Rauber, Dimitris Tsipras, Ian Goodfellow, Aleksander Madry, Alexey Kurakin.

---

### Author Response · Authors · 2023-11-23
**Thank You for the Reviews**

Dear Reviewers,

We thank the reviewers for taking the time to read and evaluate our paper. We also thank the reviewers for their valuable comments and suggestions. Reading the reviews carefully, we agree that the current manuscript lacks clarity in problem formulation and the evaluation of our claims. We thus decided to withdraw our submission from this batch for the major revisions.

Regards,
Authors